# Metabolite and Transcriptomic Changes Reveal the Cold Stratification Process in *Sinopodophyllum hexandrum* Seeds

**DOI:** 10.3390/plants13192693

**Published:** 2024-09-26

**Authors:** Rongchun Ning, Caixia Li, Tingting Fan, Tingting Ji, Wenhua Xu

**Affiliations:** 1Northwest Institute of Plateau Biology, Chinese Academy of Sciences, Xining 810001, China; ningrongchun@nwipb.cas.cn (R.N.);; 2Qinghai Key Laboratory of Qinghai-Tibet Plateau Biological Resources, Xining 810008, China; 3University of Chinese Academy of Sciences, Beijing 100049, China

**Keywords:** *Sinopodophyllum hexandrum* seed, cold stratification, dormancy, germination, plant hormones, gene expression

## Abstract

*Sinopodophyllum hexandrum* (Royle) Ying, an endangered perennial medicinal herb, exhibits morpho-physiological dormancy in its seeds, requiring cold stratification for germination. However, the precise molecular mechanisms underlying this transition from dormancy to germination remain unclear. This study integrates transcriptome and plant hormone-targeted metabolomics techniques to unravel these intricate molecular regulatory mechanisms during cold stratification in *S. hexandrum* seeds. Significant alterations in the physicochemical properties (starch, soluble sugars, soluble proteins) and enzyme activities (PK, SDH, G-6-PDH) within the seeds occur during stratification. To characterize and monitor the formation and transformation of plant hormones throughout this process, extracts from *S. hexandrum* seeds at five stratification stages of 0 days (S0), 30 days (S1), 60 days (S2), 90 days (S3), and 120 days (S4) were analyzed using UPLC-MS/MS, revealing a total of 37 differential metabolites belonging to seven major classes of plant hormones. To investigate the biosynthetic and conversion processes of plant hormones related to seed dormancy and germination, the transcriptome of *S. hexandrum* seeds was monitored via RNA-seq, revealing 65,372 differentially expressed genes associated with plant hormone synthesis and signaling. Notably, cytokinins (CKs) and gibberellins (GAs) exhibited synergistic effects, while abscisic acid (ABA) displayed antagonistic effects. Furthermore, key hub genes were identified through integrated network analysis. In this rigorous scientific study, we systematically elucidate the intricate dynamic molecular regulatory mechanisms that govern the transition from dormancy to germination in *S. hexandrum* seeds during stratification. By meticulously examining these mechanisms, we establish a solid foundation of knowledge that serves as a scientific basis for facilitating large-scale breeding programs and advancing the artificial cultivation of this highly valued medicinal plant.

## 1. Introduction

*Sinopodophyllum hexandrum* (Royle) Ying is a perennial medicinal plant belonging to the Berberidaceae family, which thrives in alpine elevations [1,2,3]. The rhizome serves as a primary source of podophyllotoxin (PPT), a metabolite with antitumor activity [4,5]. On the other hand, the dried fruit is employed in Tibetan medicine to regulate menstrual cycles and enhance blood flow [6]. Recently, it has also been recognized for its antitumor, antiviral, toxic, insecticidal, antibacterial, anti-inflammatory, antioxidant, and antiradiation activities [7,8,9,10,11,12,13]. Owing to its profound biological activities, this species has been excessively harvested over time, leading to its current classification as a second-level endangered protected plant [14,15].

*S. hexandrum*, as an esteemed medicinal plant, primarily relies on root division and seed propagation for its cultivation [16]. However, under natural conditions, the seeds of *S. hexandrum* exhibit a strong dormancy trait, with a dormancy period lasting up to ten months [17]. Research indicates that this dormancy is not attributed to a single factor but rather results from a combination of mechanical barriers caused by the seed coat and embryo, as well as the incomplete physiological maturation of the embryo, collectively known as comprehensive dormancy (morpho-physiological). This dormancy trait significantly slows down the natural renewal rate of *S. hexandrum* [18,19,20]. Cold stratification, a seed treatment technique employed to break seed dormancy and enhance germination, has proven effective in promoting the physiological maturation of *S. hexandrum* seeds [21,22]. This has been confirmed in the seeds of *Notopterygium incisum*, Angelica sinensis, and *Meconopsis punicea* [23,24,25]. It plays a significant role in facilitating the internal development and physiological maturation of the seeds, enhancing germination rates, and reducing germination latency [26,27]. Studies reveal that cold stratification triggers morphological changes in the embryo, promotes the transformation and accumulation of internal nutrients (e.g., starch, soluble protein, and sugar), and influences the content and activity of related metabolic enzymes [22,28]. These changes are interconnected and form a complex network regulated by various plant hormones. Through the fluctuations of this regulatory network, the transition of *S. hexandrum* seeds from dormancy to germination is accelerated.

Plant hormones, present at extremely low concentrations (nmol or pmol per fresh weight), function as pivotal biological signaling molecules that regulate plant growth and development, with nine known types [29,30,31,32,33]. The complex physiological processes of seed dormancy and germination involve the biosynthesis, transport, and metabolism of these hormones. Hormones such as ABA and GAs play a pivotal role in maintaining seed dormancy, while AXs and CKs promote seed germination [34]. The signaling transduction pathways of these hormones are highly interactive, forming a complex regulatory network [35]. They regulate the expression of genes related to seed dormancy and germination by activating or inhibiting specific transcription factors. The expression changes of these genes directly affect the physiological state of the seed, determining whether it transitions from dormancy to germination [36]. In the study of primary and secondary metabolites, scientists have discovered 53 compounds related to plant hormones, which are closely associated with the process of seed dormancy and germination [37]. Through complex signaling transduction pathways and gene expression regulation, these hormones jointly influence the successful germination and growth of seeds under suitable environmental conditions.

In recent years, the integration of metabolomics and other omics tools has become a prevalent approach for identifying functional genes and elucidating metabolic pathways in plants [6,38]. Here, we introduce a rapid and simultaneous LC-MS/MS method for analyzing 109 compounds, including CKs, ETH, AXs, GAs, ABAs, JAs, SAs, MLT, and SLs, during the stratification of *S. hexandrum* seeds. These analytes cover bioactive hormones, their precursors, and metabolites, aiming to capture the dynamic changes of endogenous metabolic substances. The primary objective of this study is to thoroughly investigate the key regulatory genes and their functions during the transition from dormancy to germination in *S. hexandrum* seeds under cold stratification. To accomplish this, we selected five samples from stratification stages and conducted a comprehensive analysis and comparison of the changes in plant hormone content related to seed dormancy and germination. Kyoto Encyclopedia of Genes and Genomes (KEGG) enrichment analysis and weighted correlation network analysis (WGCNA) have been employed to identify the major regulatory genes among tens of thousands of individual genes. This study utilizes a multi-omics integrated analysis to reveal the molecular mechanisms underlying the phenotypic transition from dormancy to germination in *S. hexandrum* seeds during stratification. By exploring the dynamic changes in gene expression and metabolites across different stratification stages, we aim to determine which specific genes and metabolites play crucial roles in breaking dormancy and promoting germination. Furthermore, we aim to uncover the interaction mechanisms between them and gain a deeper understanding of the reasons behind the phenotypic transition from dormancy to germination in *S. hexandrum* seeds under cold stratification. Our research not only provides a more scientific theoretical basis for the future regulation of *S. hexandrum* seed germination and conservation of germplasm resources but also explores why cold stratification treatment can significantly accelerate the transition from dormancy to germination, thereby promoting the protection and utilization of *S. hexandrum*, a second-grade endangered species.

## 2. Materials and Methods

### 2.1. Plant Material

*S. hexandrum* (Royle) Ying (Berberidaceae) seeds were collected in 2022 from the wild plant (3800–4000 m; 99°45′–101°14′ E, 32°27′–33°18′ N) in Qinghai, China. They were identified by Wenhua Xu, Associate Researcher from the Northwest Plateau Institute of Biology, Chinese Academy of Sciences. The seeds were immersed in clean water and rubbed to remove the pulp. Then, the seeds were rinsed with a large amount of purified water. After that, the shrunken seeds were sifted out and dried naturally. Finally, the seeds were stored in a refrigerator at 4 °C for future use.

### 2.2. Stratification

The sand was washed and screened (40 mesh). After draining the water, it was packed into a kraft paper bag and sterilized in a high-pressure sterilizer (103.4 kPa, 121.3 °C, 30 min). Once sterilized, the wet sand was placed in a sterilized container and left to air-dry for 40 min in a UV-lit and ventilated clean bench. The prepared seeds were soaked in 50 °C hot water for 30 min, then soaked in 1% potassium permanganate solution for 15 min to sterilize, then rinsed repeatedly with purified water, drained, and set aside for later use. A sieve (40 mesh) to be used as a stratification container was disinfected with alcohol (75%) and rinsed with purified water. A 3–4 cm layer of wet sand (moisture content: 9% to 11%) was spread at the bottom, followed by a layer of seeds and another layer of wet sand. This alternating process was repeated three times, and then the container was covered with cling film, which was punched to allow air in. It was then placed in a refrigerator (4 °C) for stratification. Every 15 days, the plastic wrap was lifted to spray purified water to maintain humidity. According to the experimental schedule, this process was repeated to layer three batches of seeds on 20 February (B1), 2 March (B2), and 16 April (B3) in 2023. Five samples were taken from these batches: 0 d (B3, S0), 30 d (B3, S1, 16 May), 60 d (B3, S2, 15 June), 90 d (B2, S3, 2 June), and 120 d (B1, S4, 20 June). Three biological replicates, each weighing 3.00 g (packed into a 2 mL plastic microtube), were flash frozen to −80 °C for the analysis of physiological and biochemical substances and transcriptome and metabolome analysis.

### 2.3. Observation of Seed Embryo Morphology and Rate

From the stratification seeds, 20 seeds were randomly selected every 15 days, soaked in purified water for 24 h (25 °C), and embedded in OCT glue for 10 min, followed by frozen embedding for 5 min at −20 °C. The Leica-CM1950 cryostat microtome was used to slice the seeds (thickness of 0.15 mm). The slices were first stained with 1% safranin solution for 3 min, then stained with 0.1% fast green alcohol solution for 30 s. The morphological structure of the seeds was observed under a Leica-DM500 microscope, and the embryo length and seed length were measured [28]. The embryo rate was calculated using the following formula:Embryo Rate (%) = (Embryo Length/Seed Length) × 100%

### 2.4. Determination of Seed Germination Rate 

We randomly selected 150 *S. hexandrum* seeds from each stratification period (0, 30, 60, 90, 120 days) and disinfected the seeds with a 1% NaClO solution for 15 min, followed by rinsing the seeds with sterile water 3–5 times. The germination experiment employed circular glass Petri dishes (d = 9 cm) as containers. Within each Petri dish, a 3–5 cm layer of sterilized sand, thoroughly moistened with sterile water, was laid as the germination bed. Fifty seeds were sown in each Petri dish, and the experiment was replicated three times. After sowing, the Petri dishes were covered and placed in a BSPX-400 GB light incubator (Shanghai, China), with the cultivation conditions set to a variable temperature mode (15 °C/25 °C) and alternating periods of darkness (10 h) and light (14 h). Each set of germination experiments lasted for 45 days. The duration of germination was determined according to the International Rules for Seed Testing; specifically, when the number of germinated seeds per day did not exceed 1% of the total number of replicate test seeds for three consecutive days, germination was considered complete. During this period, sterile water was sprayed daily to maintain the moisture of the germination bed, and the number of germinated seeds (with the emergence of the radicle as the criterion for germination) was recorded. After the experiment, the germination rate was calculated using the following formula:Germination Rate (%) = (Count of Germinated Seeds/Total Count of Test Seeds) × 100%

### 2.5. Determination of Physiological and Biochemical Substances

A total of 0.10 g of seed, after being ground and sieved on dry ice, was weighed. Using kits procured from Wuhan Nuominkeda Biotechnology Co., Ltd. (Wuhan, China), the contents of starch, soluble sugars, and soluble proteins, as well as the activities of glucose-6-phosphate dehydrogenase (G-6-PDH), pyruvate kinase (PK), and succinate dehydrogenase (SDH), were determined employing a full-wavelength microplate reader (Molecular Devices Spectra Max^®^ ABS plus, Shanghai, China) with three replicates for each measurement. We adhered to specific protocols that strictly followed the instructions provided in the respective kits.

Under heating conditions, hydrochloric acid (3 mol/L) is used to hydrolyze starch into glucose. The glucose, when it reacts with concentrated sulfuric acid, undergoes hydrolysis to produce furfural. Both the products from these reactions react with anthrone reagent to form colored complexes. These complexes exhibit their maximum absorption peaks in the visible region at 620 nm, and the optical absorption intensity is directly proportional to the content of soluble sugars and glucose in the sample. Further, the glucose content is used to calculate the starch content (mg g^−1^). Thus, by measuring the optical absorption value of the test solution at a wavelength of 620 nm, standard equations y = 2.72x + 0.05 and y = 5.56x + 0.03 are established. By substituting the obtained ΔA into these equations, x (mg mL^−1^) can be derived. Subsequently, the starch and soluble sugar content in each sample are calculated using the formulas: starch content (mg g^−1^) = 15x and soluble sugar (mg g^−1^) = 100x. Proteins, particularly cysteine and peptide bonds, possess reducing properties. When two molecules of BCA bind with Cu^+^, a purple complex is formed, which exhibits its strongest absorption peak at 562 nm. Therefore, by measuring the optical absorption value of the test solution at a wavelength of 562 nm and applying the relevant formula, the soluble protein content (mg g^−1^) can be accurately calculated [39].

G-6-PDH can reduce NADP+ to NADPH, and the change in absorbance values is measured at 340 nm over a period of 5 min (0 s and 300 s). G-6-PDH activity is calculated based on the sample mass using the formula G-6-PDH (U g^−1^) = 128.62 × (ΔA measured − ΔA blank) ÷ W (g); pyruvate kinase (PK) catalyzes the conversion of phosphoenolpyruvate into ATP and pyruvate, which is further catalyzed by lactate dehydrogenase to produce lactate and NAD+ from NADH. The rate of NADH decrease, measured at 340 nm, can be used to determine PK activity. After measuring the absorbance value at 5 s at 340 nm, the microplate containing the sample is immediately placed in the temperature-controlled area of the microplate reader and uniformly heated (maintained at 37 °C) for 10 min. The microplate is then removed, and the absorbance value is promptly measured at 10 min and 5 s. PK activity (U g^−1^) is calculated based on the sample mass using the following formula: PK activity (U g^−1^) = 1607.72 × ΔA ÷ W (g); SDH dehydrogenase catalyzes the conversion of succinate to fumarate, releasing hydrogen that is subsequently transferred by PMS to reduce Dichlorophenolindophenol (DCPIP). The reduction rate, corresponding to the absorbance difference measured at 600 nm between 10 s and 5 min, is used to calculate SDH activity. SDH activity in the sample is calculated based on the sample mass using the following formula: SDH activity (U g^−1^) = 9.234 × ΔA ÷ W (g).

### 2.6. Metabolomics Analysis of Targeted Phytohormone

A fine powder (50 mg) of freeze-dried seeds was added to 10 μL of internal standard (IS) mixed solution with a concentration of 100 ng mL^−1^ and 1 mL of methanol/water/formic acid (15:4:1, *v*/*v*/*v*) extraction agent, respectively, mixed well, and then vortexed for 10 min. Then, the homogenate was centrifuged (12,000 r/min) at 4 °C for 5 min. The supernatant was transferred into a new centrifuge tube for concentration, redissolved with 100 μL of methanol solution (80%), and then the upper portion was filtered using a dura-pore membrane (0.22 μm).

Ultra Performance Liquid Chromatography (UPLC) (ExionLC™ AD; Shanghai, China) and Tandem Mass Spectrometry (MS/MS) (QTRAP^®^ 6500^+^; Shanghai, China) were used for data acquisition. The analytical conditions were as follows: LC: column, Waters ACQUITY UPLC HSS T3 C18 (100 mm × 2.1 mm i.d., 1.8 μm); solvent system, water with 0.04% acetic acid (A), acetonitrile with 0.04% acetic acid (B); gradient program, started at 5% B (0–1 min), increased to 95% B (1–8 min), 95% B (8–9 min), finally ramped back to 5% B (9.1–12 min); flow rate, 0.35 mL/min; temperature, 40 °C; injection volume: 2 μL [40,41,42]. MS/MS: The temperature of the electrospray ionization (ESI) source is 550 °C. Under the positive ion mode, the mass spectrometry voltage is 5500 V. Under the negative ion mode, the mass spectrometry voltage is −4500 V. The curtain gas (CUR) is 35 psi. In the Q-Trap 6500^+^, each ion pair is scanned and detected according to the optimized delustering potential (DP) and collision energy (CE) [43,44]. 

The database was constructed based on standard samples for the qualitative analysis of mass spectrometry data. Data were preprocessed with Analyst 1.6.3. Standard solutions were prepared with a concentration range of 0.3–15,000 ng mL^−1^, and standard curves were plotted for detected substances using peak intensity data. Sample metabolite contents were calculated by substituting peak area ratios into the standard curve equations. UV scaling was applied to metabolite data, and HCA was conducted on their accumulation patterns using R packages. Identified metabolites were annotated using the KEGG compound database (https://www.kegg.jp/kegg/compound/ (accessed on 20 June 2024)), and annotated metabolites were then mapped to the KEGG Pathway database (https://www.kegg.jp/kegg/pathway.html (accessed on 20 June 2024)). Pathways to which significantly regulated metabolites were mapped were then fed into MSEA, and their significance was determined by the hypergeometric test’s *p*-values.

### 2.7. Differential Metabolites Selected

Differentially regulated metabolites between groups were identified by calculating the absolute Log2FC of their expression levels. Specifically, we first determined the fold change in expression levels for each metabolite between the two groups and then took the logarithm base 2 of this fold change to obtain the Log2FC. Subsequently, we conducted a rigorous screening procedure to identify metabolites exhibiting significant changes in expression levels, specifically targeting those with absolute Log2 Fold Change (Log2FC) values exceeding a pre-established threshold of |Log2FC| > 1, indicative of an expression difference of at least 2-fold between the compared conditions. Finally, to confirm that these changes in metabolites were statistically significant, we performed statistical tests (ANOVA). Through these steps, we identified those metabolites that were significantly differentially regulated between the two samples.

### 2.8. Transcriptomic Analysis

The transcriptome sequencing work was entrusted to Metware Biotechnology Co., Ltd. (Wuhan, China). RNA was extracted from 15 samples using the extraction protocol provided by the manufacturer. RNA was extracted from the samples, and its integrity, concentration, and potential DNA contamination were assessed using agarose gel electrophoresis, a Qubit 4.0 fluorometer/MD microplate reader (Shanghai, China), and a Qsep400 bioanalyzer (Shanghai, China). High-quality RNA was then selected for the construction of a cDNA library. Utilizing SBS technology, the Illumina high-throughput sequencing platform was employed to sequence the cDNA library. The resulting image data were converted into a large volume of high-quality sequencing data through CASAVA base calling. The raw sequencing data were then filtered using the fastp software (v0.23.4) and assembled using Trinity to obtain the transcript sequences of the species. Subsequently, the transcript sequences were processed with a corset to remove redundancy and obtain Unigene sequences. High-quality reads were aligned to the de-redundant transcripts to identify differentially expressed genes across different samples. Functional annotation of the Unigene sequences was achieved by utilizing DIAMOND LASTX software (v2.x) to compare them against databases such as KEGG, NR, Swiss-Prot, GO, KOG, and Trembl. This process provided valuable functional information about the genes. After predicting the amino acid sequences of the Unigenes, HMMER software (3.3.2) was employed to align them with the Pfam database, further enriching the knowledge about protein families and domains present in the samples. In summary, this comprehensive RNA sequencing and analysis pipeline offers a powerful tool for studying gene expression and functional characteristics in various species.

### 2.9. Hierarchical Cluster Analysis

The hierarchical cluster analysis (HCA) results of samples and metabolites were presented as heatmaps accompanied by dendrograms. The HCA was performed using the R package p heatmap. In this analysis, the normalized signal intensities of metabolites, which were scaled using unit variance scaling, were visualized in the heatmaps as a color spectrum ranging from low to high intensities.

### 2.10. Weighted Correlation Network Analysis

Co-expression networks were constructed using the weighted correlation network analysis (WGCNA) package in R. Eigengene values were calculated for each module and used to test associations with each hormone metabolite. For the correlation analysis of metabolites with genes, the Pearson correlation coefficient was calculated, and the plot was constructed using the R.

### 2.11. Statistical Analysis

All experiments were conducted with three independent biological replicates. Statistical analyses were performed using SPSS 27.0 software, with ANOVA and Duncan’s multiple range tests applied, with *p* < 0.05 considered to indicate a statistically significant difference between stratification stages.

## 3. Results and Discussion

### 3.1. Morphological and Phytochemical Changes during Seed Stratification

During the stratification process, significant changes occurred in the embryo morphology and embryo rate of the *S. hexandrum* seeds, as has been reported previously [18,28]. In the early stratification period (0–45 d), the embryo morphology was mostly spherical and torpedo-shaped. However, in the later stratification period (45–120 d), torpedo-shaped and cotyledonary embryos became more prevalent. So, we have selected a pair of seed morphology images in Figure 1A (representing 15 days of stratification) and Figure 1B (representing 90 d of stratification) that exhibit pronounced changes in embryo development, providing a visual illustration of the significant transformations that occur during the stratification process. As the stratification process progresses, the embryo morphology gradually transforms from globular and torpedo-shaped embryos to cotyledonary embryos, serving as a significant indicator of embryo maturation. This transition suggests that most of the seeds have completed physiological after-ripening and are capable of germination. The embryo rate of *S. hexandrum* seeds generally showed an upward trend. Specifically, there was a rapid increase in embryo rate between 75–90 d, reaching a maximum of 37% at 90 d. The germination rate of *S. hexandrum* seeds generally showed an upward trend with the extension of stratification. After 30 days of stratification, dormancy began to gradually break, and the seed germination rate increased significantly (*p* < 0.05), reaching a maximum of 47.8% after 90 days of stratification (Figure 1C). The continuous increase in embryo rate and germination rate indicates that a series of physiological and biochemical reactions have occurred within the seeds, and more seeds have completed the transition from dormancy to being ready to germinate. This change is consistent with the stratification changes observed in the seeds of *Notopterygium incisum*, *Epimedium brevicornu*, and *Acanthopanax senticosus* [24,45,46].

Figure 1D–F show the changes in the content of nutrients in the seeds of *S. hexandrum* during the cold stratification process. The starch content in the seeds generally showed a downward trend; it changed smoothly during 0–60 days and 90–120 days and decreased significantly (*p* < 0.05) after stratification to 90 days. Compared with the starch content before stratification (35.85 mg g^−1^), the starch content in the seeds at 90 d (16.67 mg g^−1^) decreased by 53.5%. The soluble protein content in the seeds showed an overall upward trend and changed smoothly during the early stratification period (0–60 d). The soluble protein content in the seeds increased significantly (*p* < 0.05) after stratification to 90 d. Compared with the soluble protein content before stratification (32.81 mg g^−1^), the soluble protein content in the seeds at 90 d (49.58 mg g^−1^) increased by 51.11%. The soluble sugar content in the seeds showed an upward trend, and the change was relatively smooth during 0–90 d, reaching the highest value (59.58 mg g^−1^) at 120 d, which was 81.59% higher than that before stratification (32.81 mg g^−1^).

The changes in the key enzyme activities of the respiratory pathway in the seeds of *S. hexandrum* during the cold stratification process are shown in Figure 1G–I. The activity of PK, a key enzyme in the respiratory pathway, generally showed a downward trend. During the initial stage of stratification (0–30 d), the enzyme activity was rapid and significantly changed (*p* < 0.05). The enzyme activity at 30 d of stratification (502.48 U g^−1^) decreased by 47.21% compared with that before stratification (739.71 U g^−1^). After that, the change was relatively stable, reaching the lowest value (387.95 U g^−1^) at 90 d of stratification, which was 48.36% lower than that before stratification (739.71 U g^−1^). The activity of G-6-PDH generally showed a slow upward trend, but the enzyme activity increased rapidly (118.63 U g^−1^) at 90 d of stratification, which was 39.32% higher than that before stratification (71.98 U g^−1^), showing a significant change (*p* < 0.05). The SDH activity showed a slow downward trend, with a significant decrease in enzyme activity during 30–60 d of stratification. The enzyme activity reached the lowest value (64.88 U g^−1^) at 120 d of stratification, which was 23.09% lower than that before stratification (89.56 U g^−1^). The results are consistent with previous studies [28], and similar changes also occur in *Corydalis tomentella* seeds, where the starch content within the seeds continues to decrease during stratification, while soluble sugar and soluble protein exhibit a trend of first decreasing and then increasing [46,47]; the activity of G-6-PDH in *Paris polyphylla* var. chinensis seeds also shows an upward trend during stratification [48].

### 3.2. The Metabolome Analysis during Seed Stratification 

#### 3.2.1. Targeted Metabolome Analysis

To investigate the underlying mechanisms of the dormancy-to-germination transition in *S. hexandrum* seeds during stratification, targeted metabolome analysis was performed using the LC-ESI-MS/MS system. Through multivariate statistical analysis, we uncovered the distribution differences of plant hormones in *S. hexandrum* seeds during stratification periods. A total of 55 metabolites were detected, providing a scientific basis for elucidating their germination mechanisms (Table 1). The identified plant hormones were displayed in a heatmap. Significant changes in the expression levels of various plant hormones were observed during different stratification stages (SM0–SM4), including a significant increase in GA5 expression in SM0, 2MeScZ (CK) in SM3, and BAP (CK) in SM4. Conversely, there was a significant decrease in the expression of IAA-Trp (Auxin) and MEJA (JA) in SM0; IAA-Phe (Auxin), Phe (SA), and GA7 in SM1; 2MeSiPR (CK) in SM3; and tz (CK) and GA9 in SM4. In subsequent data analysis, we will pay particular attention to how these significantly altered plant hormones influence the key mechanisms underlying the dormancy-to-germination transition in *S. hexandrum* seeds during stratification (Figure 2A). Compared to SM0, SM1 showed significant downregulation of 7 metabolites and upregulation of 3 metabolites; SM2 had 14 metabolites significantly downregulated and 11 significantly upregulated; SM3 exhibited 19 metabolites significantly downregulated and 7 significantly upregulated; and SM4 demonstrated 16 metabolites significantly downregulated and 7 significantly upregulated (Figure 2B). These metabolites were distinctly separated into five independent groups according to PCA analysis, indicating significant differences in the metabolites within each group (Figure 2C). During the SM0–SM4 period, there were both shared and unique patterns of plant hormone expression according to Venn analysis (Figure 2D). Cluster analysis of DEMs revealed two main clusters, with 37 distinct categories of differentially expressed metabolites (DEMs) primarily classified as ABA (1), auxin (10), CK (13), GA (6), JA (4), SA (2), and SL (1) (Figure 2E).

#### 3.2.2. The Functional Analysis of DEMs at Different Stratification Stages

To further evaluate the biological functions of DEMs at different stratification stages of *S. hexandrum* seeds, the annotation results of significantly different metabolites in KEGG are classified according to the pathway types in KEGG. When annotating the DEMs from the SM1_vs_SM0 comparison, a total of 10 KEGG pathways were identified, of which the “Ubiquinone and other terpenoid-quinone biosynthesis” pathway was highly enriched (Figure 3A). Studies have revealed that Ubiquinone (UQ) and plastoquinone (PQ) serve as crucial electron carriers in oxidative phosphorylation and photosynthesis, exhibiting a profound correlation with energy production and transformation [49]. Seeds undergo a series of intricate physiological and biochemical reactions during stratification, resulting in significant alterations in the metabolic levels of internal metabolites. We hypothesized that the significant enrichment of this metabolic pathway contributes to energy production and transfer, providing the necessary energetic support for the completion of physiological after-ripening in the seed embryo. In annotating the DEMs from the SM2_vs_SM0 comparison, we identified a total of 13 KEGG pathways. Notably, the “Plant hormone signal transduction” pathway was not only significantly enriched in the SM2_vs_SM0 comparison but also significantly enriched in SM4_vs_SM0 (Figure 3D); meanwhile, the “Carotenoid biosynthesis” and “Ubiquinone and other terpenoid-quinone biosynthesis” were highly enriched (Figure 3B). We hypothesized that the “Plant hormone signal transduction” pathway orchestrates the transition from seed dormancy to germination during stratification by modulating the balance of plant hormones, such as ABA and GA, which are pivotal regulators of dormancy and germination, in terms of their concentrations and signaling dynamics within the seed. In annotating the DEMs from the comparison, we identified a total of eight KEGG pathways. The “Carotenoid biosynthesis” in SM3_vs_SM0 was highly enriched (Figure 3C).

### 3.3. The Transcriptome Analysis during Seed Stratification 

#### 3.3.1. Transcriptome Analysis

We utilized the DIAMOND [50] BLASTX software to align the Unigene sequences against multiple databases, including KEGG, NR, Swiss-Prot, GO, KOG, and Trembl, to predict the amino acid sequences of the Unigenes. Following the completion of the prediction, we further employed the HMMER software to align the sequences against the Pfam database, aiming to acquire detailed annotation information for the Unigenes. The results annotated a total of 99,587 Unigenes (Figure 4A). We visually displayed the probability density distribution of each group’s data during the stratification stage using a violin plot that combines the characteristics of a box plot and a kernel density plot (Figure 4B). Additionally, we used the Pearson correlation coefficient (r) as an evaluation metric for biological replicate correlation and statistically analyzed the correlation between samples using a correlation plot (Figure 4C). After statistical analysis, we annotated and proofread 37,560 (37.71%) Unigenes using the SwissProt database for comparison. Simultaneously, we annotated and proofread 52,026 (52.24%) Unigenes using the TrEMBL database (Figure 4A). In the NR database comparison, a total of 52,815 Unigenes (53.3%) were annotated. Among them, 36,384 Unigenes were categorized into the top 19 identified plant species with homologous sequences, accounting for 68.9% of the total annotated Unigenes in the NR database. The remaining 31.1% of Unigenes were annotated to other plant species (Figure 4D). Through comparison and annotation with the KOG database, we successfully matched 33,542 (33.68%) Unigenes and categorized them into 25 detailed classes (Figure 4E). In terms of functional annotation, we compared the Unigenes using the GO database. The results indicated that 45,136 (45.32%) Unigenes were classified into three major categories: biological processes, cellular components, and molecular functions, which were further subdivided into 32 subcategories (Figure 4F). Finally, through comparison and annotation with the KEGG database, we annotated 40,112 (40.28%) Unigenes.

To explore the molecular mechanism of different stratification stages of *S. hexandrum* seeds, transcriptome analysis was performed. The samples of ST0 (0 d), ST1 (30 d), ST2 (60 d), ST3 (90 d), and ST4 (120 d) were collected and sequenced. After removing the low-quality reads, a total of 237,065,387 clean reads were obtained. The percentages of Q30 and GC were 95.03~95.31% and 45.04~45.99%, respectively; the high quality of the transcriptome sequencing data is evident from the metrics provided. A total of 99,587 genes were functionally annotated in the databases. Furthermore, across four distinct comparison groups, a substantial total of 65,372 significantly differentially expressed genes (DEGs) were detected. Specifically, 14,024 DEGs (8360 upregulated and 5664 downregulated), 19,137 DEGs (11,251 upregulated and 7886 downregulated), 16,501 DEGs (10,052 upregulated and 6449 downregulated), and 15,710 DEGs (9075 upregulated and 6635 downregulated) were identified in the pairwise comparisons of ST1_vs_ST0 (Figure 5A), ST2_vs_ST0 (Figure 5B), ST3_vs_ST0 (Figure 5C), and ST4_vs_ST0 (Figure 5D). To visually illustrate the overall distribution of these DEGs in the comparison groups, volcano plots were generated as follows: These DEGs were determined based on a fold change threshold of |fold change| > 2 and a corrected *p* < 0.05. During the various stages of stratification, we observed significant changes in the expression levels of differential genes in the other three stratification periods (ST1 to ST3) compared to the initial period (ST0). These changes not only include the differential genes that are co-expressed (i.e., shared) across multiple stratification stages but also those that are independently expressed (i.e., unique) at specific stratification stages. This complex expression pattern suggests that these differential genes may play specific biological functions during different stages of stratification. We systematically organized the differential gene data obtained from samples at stratification stages and plotted them into a clustering heatmap. This enabled us to visually observe significant changes in gene expression levels across different samples, clearly identify the dynamic patterns of gene expression among samples, and discern the differences and connections across various stratification periods (Figure 5E–H). These findings reveal significant changes in gene expression between different stratification stages, providing valuable insights for further research into the molecular regulatory mechanisms underlying the transition from dormancy to germination in *S. hexandrum* seeds during stratification.

#### 3.3.2. The Functional Analysis of DEGs at Different Stratification Stages

To further evaluate the biological functions of DEGs at different stratification stages of *S. hexandrum* seeds, Gene Ontology (GO) and Kyoto Encyclopedia of Genes and Genomes (KEGG) enrichment analysis of DEGs were performed. The results showed that the significantly enriched GO terms of DEGs in ST1_vs_ST0 and ST2_vs_ST0 were “kinesin complex”, “chalcone synthase activity”, and “naringenin-chalcone synthase activity” (Figure 6A,B). Among them, the most significantly enriched is the “kinesin complex” metabolic pathway. The kinesin complex, as a key molecular motor protein in cells, significantly participates in the transport process of substances within cells [51]. We speculate that this metabolic pathway exerts a significant impact on the dormancy and germination process of seeds by regulating the transport of proteins, enzymes, and hormones that are closely related to dormancy and germination. In ST3_vs_ST0, the significantly enriched GO terms among the DEGs include “extraorganismal space”, “polarity specification of adaxial axis”, and “regulation of shoot apical meristem development”. Among these, “extraorganismal space” is the most significantly enriched metabolic pathway (Figure 6C). It is directly related to the interaction between seeds and the external environment. This metabolic pathway involves the uptake of water, nutrients, and signaling molecules, providing critical fundamental substances for seed germination. We speculate that during the process of seed germination, the extra-organismal space is crucial for the absorption of external water and nutrients. Furthermore, this metabolic pathway may also be involved in the regulation of seed dormancy, where environmental signals (such as temperature and light) potentially trigger the release of dormancy by influencing metabolites in the extra-organismal space, ultimately promoting seed germination. In ST4_vs_ST0, significantly enriched GO terms among DEGs include “response to purine-containing compound”, “box C/D RNP complex”, “polarity specification of adaxial axis”, and “rRNA methyltransferase activity”. Among these, “response to purine-containing compound” was the most significantly enriched metabolic pathway (Figure 6D). “Response to purine-containing compound” is closely related to the germination and dormancy states of seeds. As a constituent of DNA and RNA, as well as an important participant in intracellular signal transduction, the normal metabolism of purines is crucial for cell division and gene expression [52].

The annotation results of significantly enriched DEGs in KEGG are classified according to the pathway types in KEGG. When annotating the DEGs from the ST1_vs_ST0 comparison, a total of 20 KEGG pathways were significantly identified, of which “Circadian rhythm-plant” was significantly enriched and “Exopolysaccharide biosynthesis” was highly enriched (Figure 6E). Circadian rhythm has a significant impact on various physiological processes in plants, including photosynthesis, stomatal opening and closing, and nutrient absorption [53]. During the stratification, the seeds of *S. hexandrum* are covered in moist sand and stored in a 4 °C constant temperature refrigerator, resulting in their prolonged isolation from oxygen and light. Based on these conditions, we speculated that the significant enrichment of the “Circadian rhythm-plant” pathway may represent a mechanism for the seeds to adapt to the stratification environment by regulating their circadian rhythm. Exopolysaccharides (EPSs) are complex polysaccharides secreted by many organisms, including plants, which play crucial roles in cell-to-cell adhesion, cell wall structure, and cell–environment interactions [54]. During stratification, a series of physiological and metabolic activities occur within the seeds. We hypothesized that the high enrichment of the EPS pathway might be a physiological response of the seeds to adapt to the stratification environment (low temperature, hypoxia, and low light). On the one hand, the increase in EPSs may help enhance the stress resistance of the seeds, protecting them from adverse environmental conditions. On the other hand, the increase in EPSs may promote information transmission and mutual recognition between cells within the seeds, creating favorable conditions for the completion of physiological maturation and germination. Annotating the DEGs from the ST3_vs_ST0 comparison, the pathway “Isoflavonoid biosynthesis” was significantly enriched, and the pathway “Exopolysaccharide biosynthesis” was highly enriched (Figure 6F). Studies have shown that the biosynthesis of isoflavones is regulated by conserved miRNAs. Given that miRNAs play a crucial role in biological processes, such as growth and proliferation, cell cycle, apoptosis, and senescence in organisms [55], we can speculate that the activation or enhancement of isoflavone biosynthesis may be closely related to the regulation of these conserved miRNAs and associated transcription factors. When annotating the DEGs from the ST4_vs_ST0 comparison, the pathway “Viral life cycle-HIV-1” was significantly enriched, and the pathway “Plant hormone signal transduction” was highly enriched (Figure 6H). Throughout the prolonged stratification process, intricate physiological alterations occur within the seeds. The remarkable enrichment of this pathway signifies the pivotal role of plant hormones in precisely regulating seed germination [56]. The hormone signal transduction machinery ensures the accurate and efficient transmission of hormonal signals, prompting the reprogramming of gene expression within the seeds. This, in turn, modulates the release of seed dormancy and initiates the germination process, offering crucial insights into the intricate mechanisms of plant hormone-mediated seed development. 

### 3.4. Metabolome and Transcriptome Integrated Analysis 

#### 3.4.1. The KEGG Combined Analysis of DEMs and DEGs

Combined and comparative KEGG enrichment analysis of DEMs and DEGs revealed 10 significantly enriched pathways in S1_vs_S0 and 8 in S3_vs_S0, with “Metabolic pathways” being the most highly enriched in both cases (Figure 7A,C). We hypothesized that the significant enrichment of this metabolic pathway indicates intense metabolic activities occurring within the seeds after 30 and 90 days of stratification. The transition from untreated to stratified conditions involves alterations in environmental factors, such as temperature, humidity, light, and oxygen levels, which prompt adjustments in the energy metabolism pathways within the seeds. These changes involve the interconversion of nutrients (starch, soluble sugars, and soluble proteins) and the activation or inhibition of metabolic pathways related to key enzymes involved in respiration, glycolysis, and the tricarboxylic acid cycle. These alterations are crucial in breaking seed dormancy and promoting germination. Comparative KEGG enrichment analysis of DEMs and DEGs revealed 13 and 9 significantly enriched pathways in S2_vs_S0 and S4_vs_S0 comparisons, with “Plant hormone signal transduction” being the most significantly enriched in both cases (Figure 7B,D). We speculate that the significant enrichment of this pathway during the 0–60 d and 0–90 d periods implies that the delicate regulation of hormone signal transduction and response, along with the modulation and balance of the interaction network among various crucial plant hormones (such as gibberellins, abscisic acid, auxin, jasmonic acid, etc.) within the seed, plays a pivotal role in breaking the dormancy state, accelerating the physiological maturation process, and ultimately facilitating the successful germination of seeds.

#### 3.4.2. Weighted Gene Co-Expression Network Analysis

To comprehensively unravel the dynamic transition of *S. hexandrum* seeds from dormancy to germination during stratification treatment, we conducted a weighted gene co-expression network analysis (WGCNA). The analysis centered on the gene expression profiles across five stratification stages (with three replicates per sample), clustering genes with similar expression patterns into distinct modules, and further delving into the intricate relationships between these modules and metabolites. Upon performing cluster analysis on all the samples, the confirmation of the absence of notable outliers indicates that the dataset possesses good consistency and clustering properties (Figure 8A). Modules are defined as highly interrelated gene clusters; genes within the same cluster have a high correlation coefficient, each tree branch constitutes a module, and each leaf represents a gene. Differences between modules are highlighted by different colors. WGCNA classifies all DEGs into 18 detailed modules (Figure 8B). By mining key information related to different modules, we have identified hub genes associated with each module (Table 2). In the stratification initial phase (ST0), a significant positive correlation is observed with yellow and pink modules, which is further found to be intimately linked to the high expression of four classes of plant hormones: ABA and IP (CK); GA1 (GAs); H2JA (JA); and JA (Figure 8C). ABA functions to inhibit seed germination, whereas GA1 and CK contribute to maintaining a dynamic dormancy–germination state by promoting cell proliferation and lifting the seed dormancy, all through the orchestration of a delicate regulatory network [57,58,59]. In the stratification phase (ST1), a significant positive correlation is observed with brown and green modules, which we found were intimately linked to the high expression of three classes of plant hormones: 2MeSiP (CK); SA and iP7G (CK); and GA6 (GAs) (Figure 8C). CKs, through their synergistic interaction with GAs, alleviate the inhibitory effect of ABA, thereby facilitating seed germination [60]. Meanwhile, GA6, a member of the gibberellin family, plays a pivotal role in breaking seed dormancy and promoting germination by upregulating the expression of genes related to cell wall degradation, metabolic processes, and transcriptional regulation, providing the necessary physiological and biochemical prerequisites for seed germination [61]. In the stratification phase (ST2), a significant positive correlation is observed with the turquoise module; we found that it was intimately linked to the high expression of two classes of plant hormones: tZRMP (CK); and GA19 and GA3 (Figure 8C). These plant hormones play a significant role in the transition from seed dormancy to germination through intricate interactions and regulatory mechanisms. In the stratification phase (ST3), a significant positive correlation is observed with red and brown modules, which we found were intimately linked to the high expression of two classes of plant hormones: iP7G (CK) and GA53 (GAs). In the stratification phase (ST4), a significant positive correlation is observed with blue, pink, and red modules; we found they were intimately linked to the high expression of GA53.

#### 3.4.3. The Co-Expression Network Analysis of DEGs and DEMs

The dormancy and germination of seeds constitute a highly complex and intricate process precisely regulated by multiple plant hormones. Our research has discovered that the seeds of *S. hexandrum* undergo a remarkable phenotypic transition from dormancy to germination during the stratification process. This transition is significantly influenced by the dynamic fluctuations and regulation of various plant hormones, including ABA, GA, IPR, 5DS, and JA. To further delve into the mechanism underlying the transition from dormancy to germination during stratification, we analyzed the KEGG pathways of differentially expressed genes and differentially accumulated metabolites and visually presented their significant correlations through a network diagram (r > 0.80, *p* < 0.05), thus uncovering the crucial molecular interplay that drives this dynamic transition. Annotating the DEGs and DEMs from the ST1_vs_ST0 comparison, the filtered network maps are “ko01100”, “ko01110”, and “ko04075” and were described as “Metabolic pathways”, “Biosynthesis of secondary metabolites” and “Plant hormone signal transduction”. These three pathways are all closely related to ABA biosynthesis (Figure 9A). Annotating the DEGs and DEMs from the ST2_vs_ST0 comparison, the filtered network maps are the same as the comparison ST1_vs_ST0; these three network maps are closely related to seven biosynthetic pathways that correlate with ABA (2), JA (1), and GA53 (4) biosynthesis (Figure 9B). Annotating the DEGs and DEMs from the ST3_vs_ST0 comparison, the filtered network maps are the same as the comparison ST2_vs_ST0; these three network maps are closely related to eleven biosynthetic pathways that correlate with ABA (3), JA (2), GA53 (2), GA3 (1), 5DS (1), and IPR (2) biosynthesis (Figure 9C). Annotating the DEGs and DEMs from the ST4_vs_ST0 comparison, the filtered network maps are “ko01100” and “ko01110”; these two network maps are closely related to eight biosynthetic pathways that correlate with ABA (1), JA (2), GA53 (1), GA3 (2), 5DS (1), and IPR (1) biosynthesis (Figure 9D).

## 4. Conclusions

This study extensively explores the physiological maturation of the embryo and the dynamic changes in internal nutrients and metabolite content in *S. hexandrum* seeds during stratification. By leveraging transcriptome and plant hormone-targeted metabolome analyses, we have uncovered the intricate molecular mechanisms governing the transition from dormancy to germination. The results demonstrate that the expression patterns of differential metabolites and genes undergo remarkable shifts across various stratification stages, which exert either positive or negative regulatory effects on modulating seed dormancy, stimulating cell proliferation, enhancing environmental adaptability, and ultimately promoting seed germination. Furthermore, the expression patterns of these differential genes align closely with the synthesis and transformation processes of plant hormones, reinforcing the essential role that hormones play in the seed germination process. Notably, hub genes have been identified, offering valuable insights for subsequent functional validation using qRT-PCR validation and refining the research framework. This comprehensive analysis ensures that the findings are not only accurate but also logically coherent, providing a robust foundation for further investigations.

## Figures and Tables

**Figure 1 plants-13-02693-f001:**
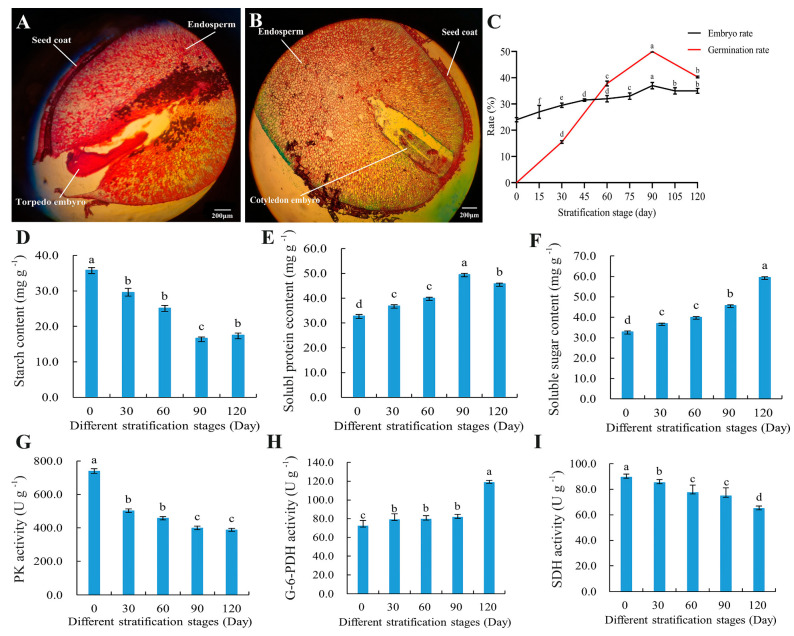
Changes in seed morphological, physiological, and biochemical features of *S*. *hexandrum* at five different stratification stages: Morphology of seed embryo (**A**,**B**). Changes in embryo rate and germination rate (**C**). Contents of soluble protein, starch, and soluble sugar (**D**–**F**). Activities of pyruvate kinase (PK), glucose-6-phosphate dehydrogenase (G-6-PDH), and succinate dehydrogenase (SDH) (**G**–**I**). Values are average with their standard deviations (*n* = 3) with three biological replicates. Different lowercase represents a significant difference (*p* < 0.05).

**Figure 2 plants-13-02693-f002:**
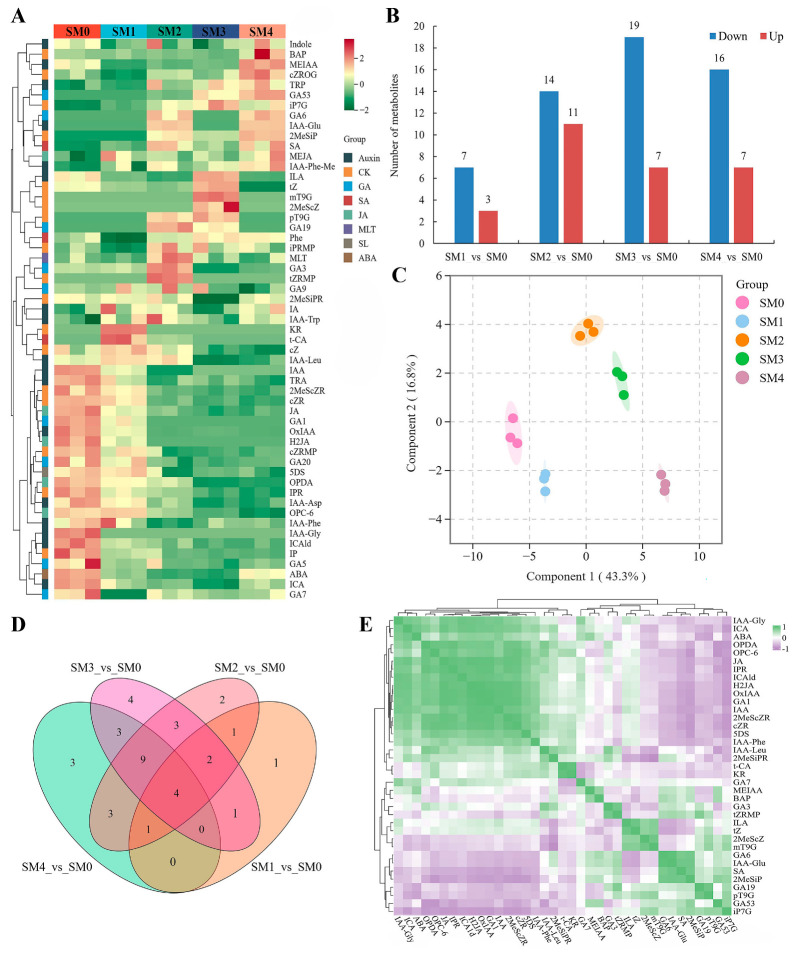
The metabolite analysis of *S. hexandrum* seeds during different stratification stages: the heatmap visualizes the total metabolites with each metabolite’s content normalized for complete linkage hierarchical clustering, where red indicates high abundance and green indicates low abundance (**A**). Bar graph analysis of total DEMs (**B**). PCA analysis of metabolites (**C**). DEMs Venn diagram (**D**). Correlation heat map between DEMs (**E**).

**Figure 3 plants-13-02693-f003:**
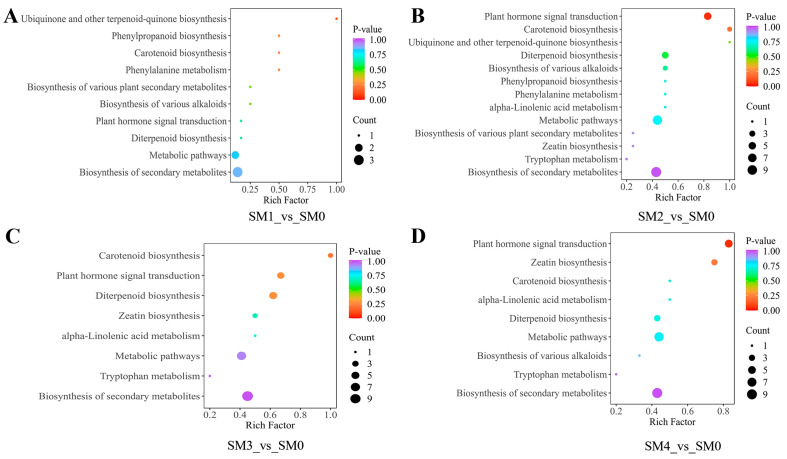
KEGG pathway analysis of DEMs in SM1_vs_SM0 (**A**). KEGG pathway analysis of DEMs in SM2_vs_SM0 (**B**). KEGG pathway analysis of DEMs in SM3_vs_SM0 (**C**). KEGG pathway analysis of DEMs in SM4_vs_SM0 (**D**). The Rich factor refers to the ratio of the number of differentially expressed genes enriched in a particular pathway to the total number of genes annotated to that pathway. A higher Rich factor indicates a greater degree of enrichment. A smaller Q-value indicates a more significant enrichment.

**Figure 4 plants-13-02693-f004:**
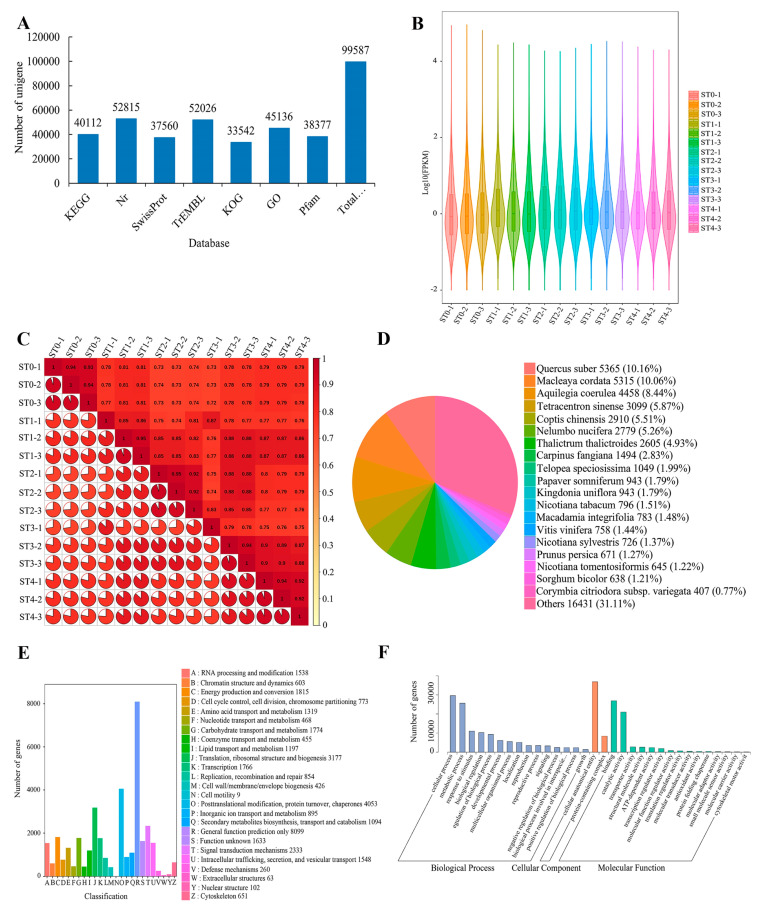
Analysis of total Unigenes and annotation status of Unigenes in various databases (**A**). Distribution and probability density display of stratified sample data (**B**). Assessment of biological replication correlation among samples using r. The closer the absolute value of r is to 1 (depicted in redder shades), the stronger the correlation (**C**). Legend shows the number of annotated orthologous clusters and genes, with different clusters represented by distinct colors (**D**). The horizontal axis represents the secondary GO terms, while the vertical axis represents the number of genes annotated to each GO term (**E**). The horizontal axis represents the functional categories of KOG IDs, while the vertical axis represents the number of genes within each category. The categories are distinguished by unique colors, and the legend provides the code and its functional description (**F**).

**Figure 5 plants-13-02693-f005:**
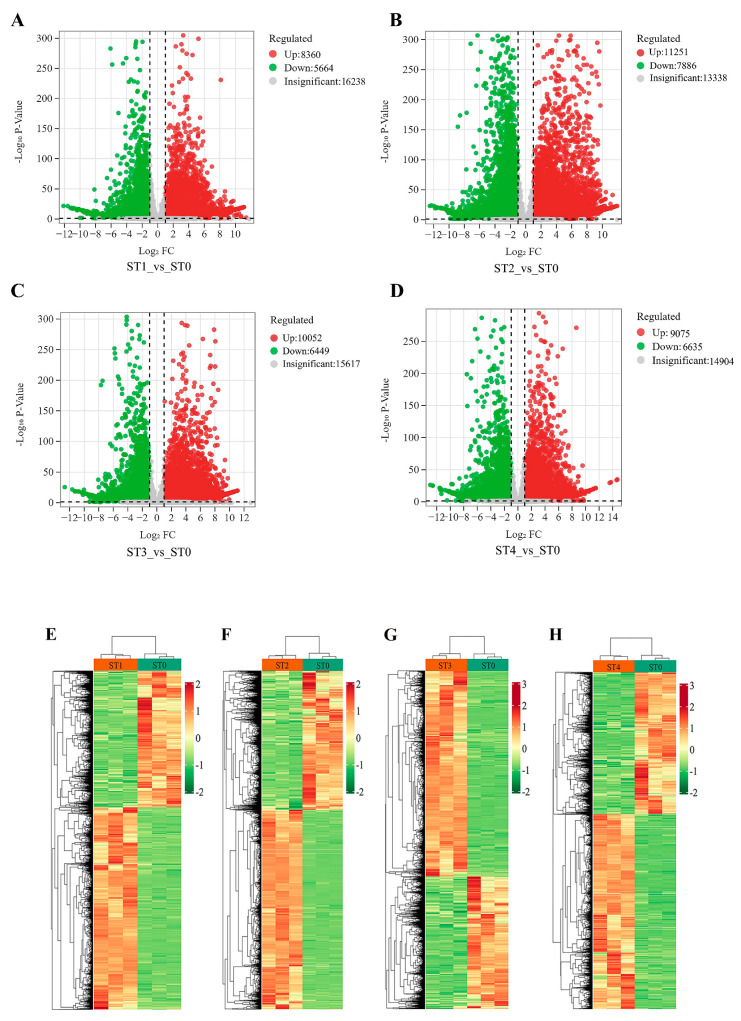
Volcano plot of differentially expressed genes between ST1_vs_ST0 (**A**), ST2_vs_ST0 (**B**), ST3_vs_ST0 (**C**), and ST4_vs_ST0 (**D**); red and green dots represent the significantly upregulated and downregulated genes. Heat map of differentially expressed genes based on hierarchical clustering analysis between ST1_vs_ST0 (**E**), ST2_vs_ST0 (**F**), ST3_vs_ST0 (**G**), and ST4_vs_ST0 (**H**) as follows: darker colors represent higher expression levels of differentially expressed genes, while lighter colors indicate the opposite.

**Figure 6 plants-13-02693-f006:**
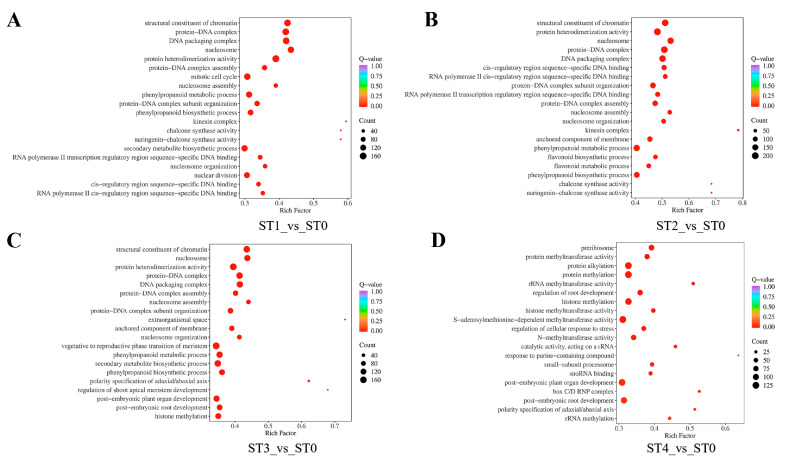
DEGs enriched on different GO terms and KEGG pathways: GO terms of DEGs in ST1_vs_ST0 (**A**). GO terms of DEGs in ST2_vs_ST0 (**B**). GO terms of DEGs in ST3_vs_ST0 (**C**). GO terms of DEGs in ST4_vs_ST0 (**D**). KEGG pathway analysis of DEGs in ST1_vs_ST0 (**E**). KEGG pathway analysis of DEGs in ST2_vs_ST0 (**F**). KEGG pathway analysis of DEGs in ST3_vs_ST0 (**G**). KEGG pathway analysis of DEGs in ST4_vs_ST0 (**H**). The Rich factor refers to the ratio of the number of differentially expressed genes enriched in a particular pathway to the total number of genes annotated to that pathway. A higher Rich factor indicates a greater degree of enrichment. A smaller Q-value indicates a more significant enrichment.

**Figure 7 plants-13-02693-f007:**
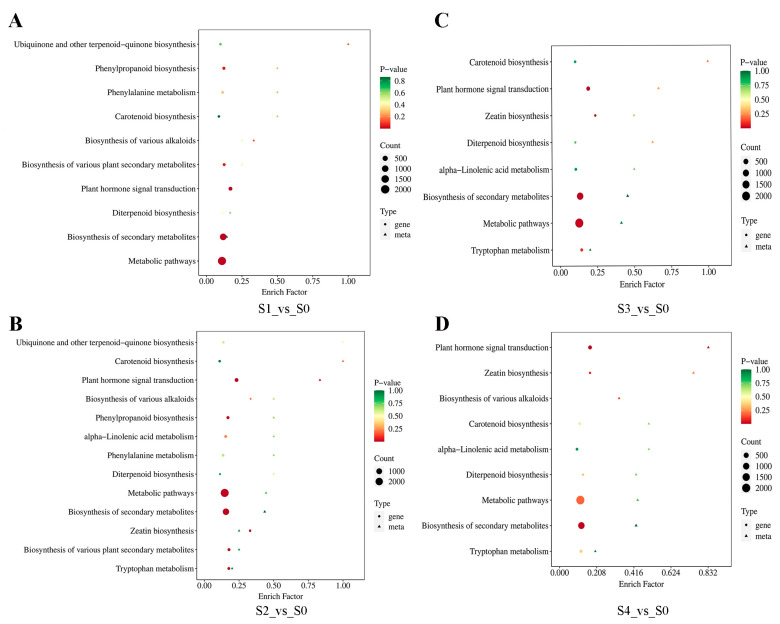
The KEGG combined analysis of DEMs and DEGs: Combined analysis of DEGs and DEMs involved in S1_vs_S0 (**A**). Combined analysis of DEGs and DEMs involved in S2_vs_S0 (**B**). Combined analysis of DEGs and DEMs involved in S3_vs_S0 (**C**). Combined analysis of DEGs and DEMs involved in S4_vs_S0 (**D**). The horizontal coordinate represents the enrichment factor of the pathway in different histologies, and the vertical coordinate represents the name of the KEGG pathway; the gradient of red-yellow-green represents the change in the significance of enrichment from high-moderate-low, indicated by *p*-value; the shape of bubbles represents different omics, and the size of the bubbles represents the number of DEMs or DEGs—the larger the number, the bigger the point.

**Figure 8 plants-13-02693-f008:**
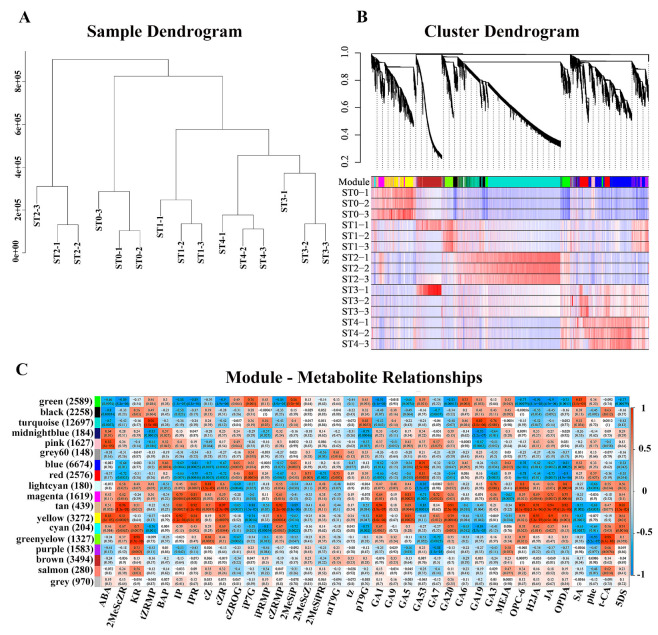
Weighted gene co-expression network analysis (WGCNA) of genes during stratification stages: Clustering dendrogram of samples based on their Euclidean distance (**A**). Hierarchical cluster tree showing co-expression modules identified by WGCNA and heat map analysis of the samples with different modules (**B**). Module–metabolite association; each row corresponds to a module, and each column represents a specific hormone (**C**). The color of each cell at the row–column intersection indicates the correlation coefficient between a module and the hormones.

**Figure 9 plants-13-02693-f009:**
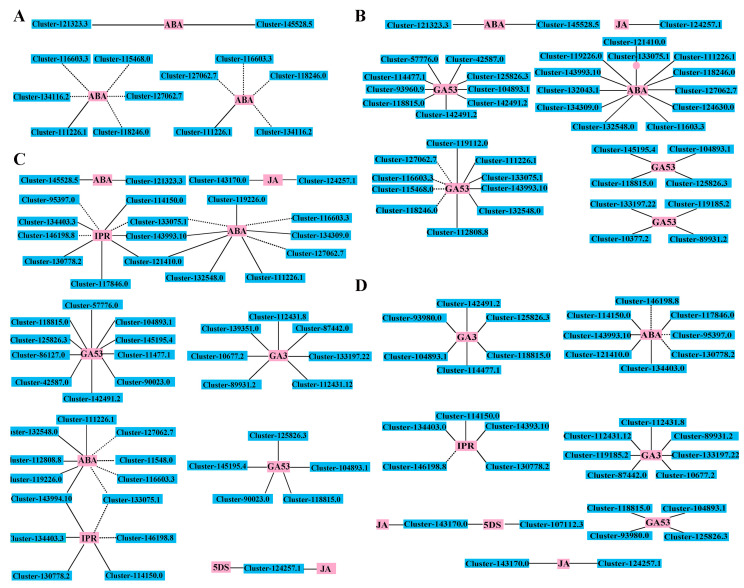
The co-expression network analysis of DEMs and DEGs based on Pearson correlation: Interaction network of DEGs and DEMs involved in ST1_vs_ST0 (**A**). Interaction network of DEGs and DEMs in ST2_vs_ST0 (**B**). Interaction network of DEGs and DEMs involved in ST3_vs_ST0 (**C**). Interaction network of DEGs and DEMs in ST4_vs_ST0 (**D**). Edges colored in pink and blue represent DEMs and DEGs, respectively; solid line and dotted line represent positive and negative correlations, The length of the lines in the network diagram does not have any practical significance. As determined by a Pearson correlation coefficient > 0.80, *p* < 0.05.

**Table 1 plants-13-02693-t001:** All metabolites detected from 15 samples.

Abbreviation	Metabolite	Class
mT9G	Meta-Topolin-9-glucoside	Cytokinin
tZRMP	9-Ribosyl-trans-zeatin 5′-monophosphate	Cytokinin
2MeScZR	2-Methylthio-cis-zeatin riboside	Cytokinin
KR	Kinetin riboside	Cytokinin
tZ	Trans-Zeatin	Cytokinin
cZRMP	Cis-Zeatin riboside monophosphate	Cytokinin
IPR	N6-isopentenyladenosine	Cytokinin
2MeSiPR	2-Methylthio-N6-isopentenyladenosine	Cytokinin
BAP	6-Benzyladenine	Cytokinin
2MeSiP	2-Methylthio-N6-isopentenyladenine	Cytokinin
iP7G	N6-Isopentenyl-adenine-7-glucoside	Cytokinin
iPRMP	N-6-iso-pentenyladenosine-5′-monophosphate	Cytokinin
cZR	Cis-Zeatin riboside	Cytokinin
IP	N6-isopentenyladenine	Cytokinin
pT9G	4-[[(9-beta-D-Glucopyranosyl-9H-purin-6-yl) amino] methyl] phenol	Cytokinin
2MeScZ	2-Methylthio-cis-zeatin	Cytokinin
cZROG	Cis-Zeatin-O-glucoside riboside	Cytokinin
cZ	cis-Zeatin	Cytokinin
IAA-Gly	Indole-3-acetyl glycine	Auxin
IAA-Glu	Indole-3-acetyl glutamic acid	Auxin
IA	3-Indoleacrylic acid	Auxin
IAA-Leu	N-(3-Indolylacetyl)-L-leucine	Auxin
IAA-Phe	N-(3-Indolylacetyl)-L-phenylalanine	Auxin
IAA-Trp	Indole-3-acetyl-L-tryptophan	Auxin
ILA	Indole-3-lactic acid	Auxin
OxIAA	2-oxindole-3-acetic acid	Auxin
TRP	L-tryptophan	Auxin
IAA	Indole-3-acetic acid	Auxin
TRA	Tryptamine	Auxin
MEIAA	Methyl indole-3-acetate	Auxin
Indole	Indole	Auxin
ICAld	Indole-3-carboxaldehyde	Auxin
ICA	Indole-3-carboxylic acid	Auxin
IAA-Phe-Me	Indole-3-acetyl-L-phenylalanine methyl ester	Auxin
IAA-Asp	Indole-3-acetyl-L-aspartic acid	Auxin
GA9	Gibberellin A9	Gibberellin
GA53	Gibberellin A53	Gibberellin
GA7	Gibberellin A7	Gibberellin
GA3	Gibberellin A3	Gibberellin
GA6	Gibberellin A6	Gibberellin
GA19	Gibberellin A19	Gibberellin
GA1	Gibberellin A1	Gibberellin
GA5	Gibberellin A5	Gibberellin
GA20	Gibberellin A20	Gibberellin
OPC-6	3-oxo-2-(2-(Z)-Pentenyl) cyclopentane-1-hexanoic acid	Jasmonic acid
MEJA	Methyl jasmonate	Jasmonic acid
JA	Jasmonic acid	Jasmonic acid
OPDA	Cis (+)-12-Oxophytodienoic acid	Jasmonic acid
H2JA	Dihydrojasmonic acid	Jasmonic acid
SA	Salicylic acid	Salicylic acid
Phe	L-Phenylalanine	Salicylic acid
t-CA	Trans-Cinnamic acid	Salicylic acid
ABA	Abscisic acid	Abscisic acid
MLT	Melatonine	Melatonin
5DS	5-Deoxystrigol	Strigolactone

**Table 2 plants-13-02693-t002:** Hub genes of each module.

Module	Hub Genes	Module	Hub Genes
Black	Cluster-103188.0	Magenta	Cluster-140764.5
Blue	Cluster-16348.6	Mid night blue	Cluster-141210.0
Brown	Cluster-3253.1464	Pink	Cluster-135750.4
Cyan	Cluster-123911.19	Purple	Cluster-127961.0
Green	Cluster-132019.0	Red	Cluster-139658.0
Green yellow	Cluster-85301.32	Salmon	Cluster-57738.15
Grey60	Cluster-4422.225	Tan	Cluster-138313.7
Light cyan	Cluster-114960.0	Turquoise	^1^ Cluster-112157.0
Yellow	Cluster-112291.0		

^1^ Cluster refers to a group formed by a series of related transcripts that are grouped together. These transcripts may originate from the same gene, but due to splicing variations, post-transcriptional modifications, or other reasons, they may have different sequences.

## Data Availability

The data presented in this study are available on request from the corresponding author. The data are not publicly available due to privacy.

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
