# Peer review of "Metabolite and Transcriptomic Changes Reveal the Cold Stratification Process in Sinopodophyllum hexandrum Seeds"

_plants, 2024, doi:10.3390/plants13192693_

Round 1
Reviewer 1 Report
Comments and Suggestions for Authors
Comments to the manuscript
Metabolite and transcriptomic changes reveal the cold stratifi-2 cation process in Sinopodophyllum hexandrum seed by Ning et al. submitted to Plants.
The idea of the manuscript is very interesting, as there is a lack of such data in the literature. So the novelty of the work is high, and the manuscript could be o high interest for researchers working with seeds. But after reading a part of the manuscript ( pages 1-7) my opinion is that in the current form it cannot be accepted for publication. There are a lot of methodological shorcomings, that needs clarification. Therefore I decided to finish my review at that stage, and ask authors for improvement of M&M - because correctness of the methods is crucial for data presentation and analysis.
Detailed comments
Introduction: line 51 - cold stratificiation is not an artifical metod for dormancy removal, it is a treatment, and it occuers in natural condition in temperature climate - please modify.
line 63-69 - such information are obvious, should be shortened
line 93 - modify the grammmar, it is future time used, but your work has been done.
Material and Methods
This is not the laboratory guidelines, so please use a proper grammar mode that is needed in scientific papers. Modify all over the M&M section.
Please explain this step: The prepared seeds were soaked in 50℃ hot water for 30 minutes - the temperature is high and the duration of the treatment is quite long - what is the reason for this step?
What was the moisture content of sand used for stratification treatment - include information on sand and water weight.
line 136, please explain the longevity of seed soaking - 24 hours (25 ℃) - was it germination period, culture... This is incomprehensible.
What about germination rate of the seeds after stratification for 30, 60, 90, 120 days?
Determination of Physiological and Biochemical Substances - needs detailed information on the weight of the seeds, volume of the extraction buffer etc. Information that " it was done in accordance with the specific instructions provided for each respective kit" is unacceptable. The comments refers to all the paragraph.
Include detailed information on (UPLC - MS/MS) equipment used in analysis.
Transcriptomic analysis - include information on RNA extraction.
As I have realized seeds of Sinopodophyllum hexandrum are characterized by morpho-physiological dormancy but not physiological dormancy as was mentioned by the authors in the Abstract. At the beginning of the manuscript it is very confusing. Information on germination rate after 30, 60, 90, 120 days of stratification are necessary - please include such data.
Results: e.g. 1 A, B - what is the stage of stratification? Why are only two photos included? I would expected photographs after 30, 60, 90 and 120 days of stratification to compare maturation of the seed - thus the presented data are incoplete.
I would also suggest to cite some papers related to seed stratification and dormancy from Baskin & Baskin team .
Comments on the Quality of English LanguageThe grammar mode of M&M section is not appropiate for publication. Authors have used a mode typical for lab. recipe. It needs rewritting.
Author Response
Response to Reviewer's Comments
Response to Reviewer 1 |
||
On behalf of my co-authors, we express our profound gratitude for your meticulous and professional review of our article. We sincerely acknowledge that your insightful feedback has pinpointed several areas where we can further refine and improve our work. Furthermore, we are deeply appreciative of your positive and constructive comments, which have been incredibly valuable in enhancing the quality of our research. |
||
1. Questions for General Evaluation |
Reviewer’s Evaluation |
|
Does the introduction provide sufficient background and include all relevant references? |
Yes |
|
Is the research design appropriate? |
Must be improved |
|
Are the methods adequately described? |
Must be improved |
|
Are the results clearly presented? |
Can be improved |
|
Are the conclusions supported by the results? |
Yes |
|
3. Point-by-point response to reviewer Introduction |
||
Comments 1: line 51 - cold stratification is not an artificial method for dormancy removal, it is a treatment, and it occurs in natural condition in temperature climate - please modify. |
||
Response 1: We sincerely thanks for your careful reading, we agree with this comment. As suggested by you, we have changed the “an artificial method” to “a seed treatment technique.” The complete and revised sentence is “Cold stratification, a seed treatment technique employed to break seed dormancy and enhance germination, has proven effective promoting the physiological maturation of S. hexandrum seeds.” You can find the modifications I have made on lines 51-53 of page 2 of the revised manuscript I submitted. |
||
Comments 2: line 63-69 - such information are obvious, should be shortened |
||
Response 2: Thanks for your suggestion. we agree with your comment. As suggested by you, we have shortened the sentence: “Plant hormones serve as crucial biological signaling molecules regulating plant growth and development with extremely low concentrations (nmol or pmol per fresh weight). Nine major categories of plant hormones have been identified, including cytokinins (CKs), gibberellins (GAs), ethylene (ETH), abscisic acid (ABA), auxin (AXs), brassinosteroids (BRs), jasmonic acid (JA), salicylic acid (SA), and strigolactones (SLs).” to “Plant hormones, present at extremely low concentrations (nmol or pmol per fresh weight), function as pivotal biological signaling molecules, regulating plant growth and development, with nine known types.” You can find the modifications I have made on lines 63-65 of page 2 of the revised manuscript I submitted. |
||
Comments 3: line 93 - modify the grammar, it is future time used, but your work has been done. Response 3: Thanks for your suggestion. we agree with your comment. As suggested by you, we have changed the future tense to present perfect tense. Specifically, we have changed “will be employed” to “have employed.” You can find the modifications I have made on page 2, paragraph 3, line 91 of the revised manuscript I submitted. Material and Methods: Comments 1: Please explain this step: The prepared seeds were soaked in 50℃ hot water for 30 minutes - the temperature is high and the duration of the treatment is quite long - what is the reason for this step? Response 1: We sincerely appreciate your meticulous review and have provided detailed responses to your inquiries. The reasons for adopting the hot water seed disinfection method (50°C soaking for 30 minutes) for stratified seeds are as follows: a. The hot water treatment is primarily designed to significantly reduce the initial population of mold spores and other potential pathogens adhering to the seed surface, thereby mitigating the risk of subsequent mold development. b. In the selection process of stratified seeds, we rigorously adhere to established screening criteria and conduct viability tests to ensure seed quality. Nevertheless, the inadvertent inclusion of unhealthy seeds remains inevitable. For these seeds, stratification treatment not only fails to enhance physiological maturity but may expedite mold growth. Hot water soaking enables the removal of unhealthy (dead) seeds that float to the surface, effectively decreasing the mold load within the stratification environment and preventing further mold contamination of seeds. c. The experiments were conducted in February (average temperature: −11°C- 4°C), March (average temperature: −4°C-10°C), and April (average temperature: 2°C -11°C) of 2023 in Xining, Qinghai Province, the average temperatures were relatively low. Thus, the 50°C water used for soaking was not maintained at a constant temperature throughout the 30 min. Given the significant temperature differential with the ambient environment, the water cooled rapidly, resulting in an actual soaking time at 50°C of approximately 10-15 minutes. d. Hot water soaking not only eliminates mold attached to the seed surface but also stimulates seed germination. This method is commonly employed as a pretreatment for chili and rice seeds, among others. Comments 2: What was the moisture content of sand used for stratification treatment - include information on sand and water weight. line 136, please explain the longevity of seed soaking - 24 hours (25 ℃) - was germination period, culture... This is incomprehensible. Response 2: we agree with this comment. Thank you for raising these questions, which have helped me refine my work. Here is the answer to your first question: The moisture content of the sand used for stratification is 9% to 11%. We replaced “able to form a ball when squeezed and disperse easily when gently pressed” with “moisture content: 9% to 11%”. You can find the modifications I have made on page 3, paragraph 2, line 122 of the revised manuscript I submitted. Next is the answer to your second question: a. Softening Seeds: Soaking can soften the shells and outer tissues of seeds, facilitating subsequent slicing. b. Reducing Brittleness: Soaking can make the seeds more pliable, reducing brittleness during slicing or preparation. c. Enhancing Staining: After slicing, the seeds need to be stained to observe the embryo and endosperm, and soaking can help the staining solution better penetrate into the interior of the seeds. d. Preventing Sample Drying: In electron microscope experiments, samples need to maintain a certain level of moisture. Soaking can prevent the seeds from drying out excessively during preparation and observation. Comments 3: What about germination rate of the seeds after stratification for 30, 60, 90, 120 days? Response 3: Thanks for your suggestions on our research. In my opinion, the core objective of this study is to delve deeply into the physicochemical properties, metabolite changes, and the molecular mechanisms regulating genes during the stratification process of S. hexandrum seeds. Recognizing the significance of stratification in completing the physiological maturation of seeds, we paid particular attention to the measurement and analysis of seed embryo rates. During the stratification process, we observed that some seeds began to naturally germinate (exhibit a white radicle) starting from 60 d. To ensure representativeness and randomness in our germination experiments, we adopted a principle of random sampling. This led to a challenge: whether or not to include these naturally germinated seeds in the calculation of germination rates. Based on this challenge, we initially decided not to include the germination rate data in our final results. However, heeding your advice, we have now supplemented the germination experiment data (excluding naturally germinated seeds) into Figure 1C. You can find the modifications I have made on page 7, paragraph 3, line 327-331 and 337-340 of the revised manuscript I submitted. At the same time, relevant content related to the germination experiment has been added to both the Materials and Methods section as well as the Results section of the manuscript. Allowing you and other readers to evaluate our research findings more comprehensively. You can find the modifications I have made on page 3, paragraph 4, line 143-161 of the revised manuscript I submitted. Comments 4: Determination of Physiological and Biochemical Substances - needs detailed information on the weight of the seeds, volume of the extraction buffer etc. Information that “it was done in accordance with the specific instructions provided for each respective kit" is unacceptable. The comments refer to all the paragraph. Response 4: Thanks for your suggestion. we agree with your comment. As suggested by you, we have added the detailed information about weight of the seeds, volume of the extraction buffer etc. Specifically, we have supplemented in detail the detection principles, detection methods, operation procedures and relevant references of three nutrients, as well as the detailed detection methods of the activity of three key enzymes. You can find the modifications I have made on page 4, paragraph 1-7, line 162-239 of the revised manuscript I submitted. Comments 5: Include detailed information on (UPLC - MS/MS) equipment used in analysis. Response 5: Thanks for your suggestion. we agree with your comment. As suggested by you, we have added the detailed information about equipment. Specifically, we have added includes the model of the equipment, the dosages of various reagents, mobile phases used in UPLC, the chromatographic column, the gradient elution program, flow rate, column temperature, and injection volume. For MS/MS detection, the relevant detailed information encompasses the temperature and voltage of the electrospray ionization source, as well as the curtain gas and other parameters and the references that are closely related to these conditions. You can find the modifications I have made on page 5, paragraph 5, line 247-258 of the revised manuscript I submitted. And 42-46 references were added. You can find the modifications I have made on page 25, line 877-887 of the revised manuscript I submitted. Comments 6: Transcriptomic analysis - include information on RNA extraction. Response 6: Thanks for your suggestion. we agree with your comment. As suggested by you, we have added the information on RNA extraction. The transcriptome sequencing of the 15 samples was entrusted to Metware Biotechnology Co., Ltd. (Wu han, China) for detection, so we added “The transcriptome sequencing work is entrusted to Metware Biotechnology Co., Ltd. (Wu han, China).”to the manuscript. Meanwhile we have replaced “RNA was extracted from the samples” with “RNA was extracted from 15 samples using the extraction protocol provided by the manufacturer.” You can find the modifications I have made on page 6, paragraph 3, line 283-285 of the revised manuscript I submitted. Comments 7: As I have realized seeds of S. hexandrum are characterized by morpho-physiological dormancy but not physiological dormancy as was mentioned by the authors in the Abstract. At the beginning of the manuscript it is very confusing. Response 7: Thank you very much for your thorough review and valuable comments on our research. You have pointed out the incorrect statement in the abstract, where the dormancy type of S. hexandrum seeds was mistakenly described as physiological dormancy. We sincerely apologize for this and would like to clarify and correct it. In preparing the abstract, due to my oversight, I failed to accurately distinguish between the concepts of "physiological dormancy" and "morpho-physiological dormancy." In fact, as you keenly observed in your review, the dormancy type of S. hexandrum seeds is morpho-physiological dormancy, where the dormancy of S. hexandrum is not only regulated by internal physiological mechanisms but also constrained by the seed coat and endosperm. To eliminate this confusion, we have revised the abstract as follows: “Sinopodophyllum hexandrum, an endangered perennial medicinal herb, exhibits morpho-physiological dormancy in its seeds, requiring cold stratification for germination.” Additionally, we have carefully reviewed the entire manuscript to ensure consistency and accuracy in the description of the dormancy characteristics of S. hexandrum seeds throughout the manuscript. So, we have added “morpho-physiological” to introduction section. Once again, thanks for your attention to our research and for your insightful corrections. Your valuable feedback will play a crucial role in refining our study and enhancing the quality of our paper. You can find the modifications I have made on page 1, line 10 and page 2, line 50 of the revised manuscript I submitted. Comments 8: Information on germination rate after 30, 60, 90, 120 days of stratification are necessary - please include such data. Response 8: Thank you for highlighting the importance of the germination rate data during the stratification process of S. hexandrum seeds, which has allowed me to correct the oversight of prioritizing only the embryo rate data. We have now presented the germination-related data in Figure 1(C), and added relevant information in both the Materials and Methods section as well as the Results section. You can find the modifications I have made on page 3, paragraph 4, line 143-161 and page 7, paragraph 3, line 337-340 of the revised manuscript I submitted. Comments 9: Results: e.g. 1 A, B - what is the stage of stratification? Why are only two photos included? I would expect photographs after 30, 60, 90 and 120 days of stratification to compare maturation of the seed - thus the presented data are incomplete. Response 9: Thank you very much for your thorough review and valuable comments on our research. You have pointed out the unclear statement in the Results。We sincerely apologize for any ambiguity in our previous statement and would like to provide a clearer explanation to rectify it. During the stratification experiment, we selected seeds every 15 days for electron microscopy examination and statistical data collection. Based on the obtained data, we plotted a graph of embryo rate over time. From this graph, we observed a gradual increase in embryo rate during the initial 0-30 days, which led us to include Figure 1A, representing the seed embryo morphology during the early stages of stratification. Additionally, we noticed a more significant change in embryo rate between 75-90 days of stratification, hence the decision to showcase Figure 1B, depicting the embryo morphology at 90 days. These two figures sufficiently and intuitively illustrate the notable transformations that occur in the seed embryo during the stratification process. It is worth noting that the morphological changes in the seed embryo during stratification are not as pronounced or straightforward as those observed in fruit ripening or direct germination. As evident from the graph, the embryo rate changes are relatively gradual, with no discernible change during 45-60 days of stratification, and even a slight decline from 90-120 days. Taking these factors into consideration, we have chosen not to exhaustively present the embryo morphology at every interval (i.e., 30, 60, 90, 120 days). Instead, we have leveraged metabolome and transcriptome data to provide a direct visualization of the expression intensities of differential metabolites and the up- or down-regulation of genes across various stages of stratification. This approach offers a comprehensive understanding of the underlying mechanisms and key factors influencing the stratification process. We have corrected and clarified in the corresponding section of the manuscript the specific stratification stages that figure A and B represents. Specifically, we have added “So, we have selected a pair of seed morphology images, Figure 1A (representing 15 days of stratification) and Figure 1B (representing 90 days of stratification), that exhibit pronounced changes in embryo development, providing a visual illustration of the significant transformations that occur during the stratification process.” to corresponding part. You can find the modifications I have made on page 3, paragraph 4, line 143-161 and page 7, paragraph 3, line 327-331 of the revised manuscript I submitted. |

Reviewer 2 Report
Comments and Suggestions for Authors
the research topic is exciting and the authors fulfilled their framework. the introduction is very informative with enough information about similar research. the introduction is obvious with a focus on pointing out the importance of the Sinopodophyllum hexandrum, its characteristics, and its uses in traditional medicine. The references are carefully selected and add value to the research. the materials and methods give enough information for the reader to repeat experiments.
the results are presented well pointing out the influence of the stratification stages to the germination and plants composition.. the presented figures gives paper additional value and make paper more understandable. the discussion is appropriate and well-analyzed. the conclusion summarizes findings and underlines the most important one.
but I would suggest authors to better explain the descriptive statistics of the method, i.e. when there is the determination of some analite the paper should have the description of the method's accuracy, precision, and recovery.
in this paper, I don't see anything of that and data are presented through graphs only hence the tables could improve it.
Comments on the Quality of English Languagethe paper is easy to read and understand hence the English style and grammar are satisfactory
Author Response
Response to Reviewer's Comments
Response to Reviewer 2 |
||
On behalf of my co-authors, we express our profound gratitude for your meticulous and professional review of our article. We sincerely acknowledge that your insightful feedback has pinpointed several aspects where we can further refine and improve our work. Additionally, we are deeply honored by your recognition of our research topic and our efforts, as well as your positive evaluation of the paper's structure and content. We truly appreciate your positive and constructive comments, which are invaluable in enhancing the quality of our research. |
||
1. Questions for General Evaluation |
Reviewer’s Evaluation |
|
Does the introduction provide sufficient background and include all relevant references? |
Yes |
|
Is the research design appropriate? |
Can be improved |
|
Are the methods adequately described? |
Can be improved |
|
Are the results clearly presented? |
Yes |
|
Are the conclusions supported by the results? |
Yes |
|
Material and Methods: Comments 1: I would suggest authors to better explain the descriptive statistics of the method, i.e. when there is the determination of some analyte the paper should have the description of the method's accuracy, precision, and recovery. |
||
Response 1: We sincerely thanks for your careful reading, we agree with your comment. As suggested by you, we have made notable improvements to the Materials and Methods section by providing more precise and detailed descriptions of various methods, aiming to enhance the accuracy and reproducibility of the methods and ensure the reliability and verifiability of the research results. We have specifically made the following improvements: a. Initially, our description of the sand moisture during the stratification process was primarily based on empirical judgments. To significantly enhance the rigor of our research and the reproducibility of experiments, we made an improvement: we replaced “able to form a ball when squeezed and disperse easily when gently pressed” with “moisture content: 9% to 11%”. This precise modification not only clearly defines the water content of the sand but also provides a quantifiable standard for subsequent experiments, thereby greatly enhancing the reproducibility of experiments and the reliability of results. You can find the modifications I have made on page 3, paragraph 2, line 122 of the revised manuscript I submitted. b. We have supplemented in detail the detection principles, detection methods, operation procedures and relevant references of three nutrients, as well as the detailed detection methods of the activity of three key enzymes. This enhancement significantly strengthens the scientific rigor and accuracy of our research, while also laying a solid foundation for subsequent studies by improving reproducibility. You can find the modifications I have made on page 4, paragraph 1-7, line 163-239 of the revised manuscript I submitted. c. We have added the detailed information about (UPLC-MS/MS) equipment used in analysis. we have added includes the model of the equipment, the dosages of various reagents, mobile phases used in UPLC, the chromatographic column, the gradient elution program, flow rate, column temperature, and injection volume. For MS/MS detection, the relevant detailed information encompasses the temperature and voltage of the electrospray ionization source, as well as the curtain gas and other parameters and the references that are closely related to these conditions. You can find the modifications I have made on page 5, paragraph 5, line 247-258 of the revised manuscript I submitted. And 42-46 references were added. on page 25, line 877-887 of the revised manuscript I submitted. d. We have added the information on RNA extraction. The transcriptome sequencing of the 15 samples was entrusted to Metware Biotechnology Co., Ltd. (Wu han, China) for detection, so we added “The transcriptome sequencing work is entrusted to Metware Biotechnology Co., Ltd. (Wu han, China).”to the manuscript. Meanwhile we have replaced “RNA was extracted from the samples” with “RNA was extracted from 15 samples using the extraction protocol provided by the manufacturer.” You can find the modifications I have made on page 6, paragraph 3, line 283-285 of the revised manuscript I submitted. |
||
Results and Discussion: Comments 2: I don't see anything of that and data are presented through graphs only hence the tables could improve it. Response 2: We sincerely thanks for your careful reading, we agree with your comment. As suggested by you, we have added Table 1 to part 3.4.2. Weighted gene co-expression network analysis. The title of the table is “Hub genes of each module”. Relevant descriptions have been added to the article. You can find the modifications I have made on page 19, line 654-655 and page 20, line 687-691 of the revised manuscript I submitted. Table 1. Hub genes of each module |
Module |
Hub Genes |
Module |
Hub Genes |
|
Black |
Cluster-103188.0 |
Magenta |
Cluster-140764.5 |
|
Blue |
Cluster-16348.6 |
Mid night blue |
Cluster-141210.0 |
|
Brown |
Cluster-3253.1464 |
Pink |
Cluster-135750.4 |
|
Cyan |
Cluster-123911.19 |
Purple |
Cluster-127961.0 |
|
Green |
Cluster-132019.0 |
Red |
Cluster-139658.0 |
|
Green yellow |
Cluster-85301.32 |
Salmon |
Cluster-57738.15 |
|
Grey60 |
Cluster-4422.225 |
Tan |
Cluster-138313.7 |
|
Light cyan |
Cluster-114960.0 |
Turquoise |
1Cluster-112157.0 |
|
Yellow |
Cluster-112291.0 |
1Cluster: refers to a group formed by a series of related transcripts that are grouped together. These transcripts may originate from the same gene, but due to splicing variations, post-transcriptional modifications, or other reasons, they may have different sequences.

Round 2
Reviewer 1 Report
Comments and Suggestions for Authors
Comments to revised manuscript "Metabolite and transcriptomic changes reveal the cold stratification process in Sinopodophyllum hexandrum seed" by Ning et al. submitted to Plants.
The authors improved partially the manuscript according to my suggestions. Material and methods section has been completed but still needs revision. It is written as a recipe but not the description of M&M - it needs rewriting. In general all over the text of M&M needs English improvement. I have detected a huge gap between description of M&M vs. Results and Discussion - probably because these parts were written by two persons - please uniform style.
The manuscript is interesting and presents interesting and valuable data, although the discussion of the data is very brief, very general and does not contribute too much to our knowledge. To sum up, the presented data are not very well interpreted - interpretation should be improved. Conclusion is very general, even too universal - may fit to many data, not exactly presented in the manuscript.
Detailed comments are in the pdf version of the manuscript.

Material and methods section needs great improvement, there are some mistake all over the text - needs revision.
Author Response
3. Point-by-point response to reviewer Materials and Methods |
Point 1: line 164 -240: Please modify all over the text the mode. This is the lab. recipe, but not the description for M&M |
Response 1: We sincerely thanks for your careful reading, we agree with your comments. This part of work was done in my early days, my description of specific experimental procedures was somewhat unpolished. To rectify this, I downloaded relevant papers from Plants, studied the way experiments are described, and thoroughly reviewed the instructions for the reagent kits. Furthermore, I incorporated your invaluable suggestions from the two rounds and conducted a comprehensive revision and enhancement of the "2.5. Determination of Physiological and Biochemical Substances" I sincerely hope that the modifications and upgrades made this time will be of substantial significance. To enhance the precision and logicality of the experiment, we have optimized the experimental procedure and supplemented detailed information regarding the nutrients, enzyme assays, the weight, the detection instruments, detection principles, reagents used, their respective units and formula. Furthermore, we have refined and condensed the language also made corresponding modifications to the title of the y-axis in Figure 1. You can find the modifications I have made on page 4, paragraph 1, line 162-201 and page 8, line 341 of the revised manuscript and Figure I submitted. |
Point 2: line 344: The description is inaccurate.
Response 2: We sincerely thanks for your careful reading, we agree with this comment. As suggested by you, we have changed the “release” to “being ready to germinate”. You can find the modifications I have made on page 6, paragraph 4, line 305 of the revised manuscript I submitted.
Point 3: The unit representation is inappropriate all over the text and at Fig. Include information on the unit into M&M description, specify DW or FW use all over the text and at Fig.
Response 3: We sincerely thanks for your careful reading, we agree with this comment. As suggested by you, we have changed the “mg/g” to “mg g -1”. All unit expressions throughout the entire article have been revised. You can find the modifications made are highlighted throughout the entire article. Regarding your second question, the precise definition of the seed state used for physicochemical property testing, I have the following concerns. During the stratification process, seeds are exposed to simulated winter conditions, stored in a low-temperature environment under a covering of moist sand for 1 - 4 months. Compared to fresh samples, these stratified seeds have lost some of their free water content, yet they have not reached the full dryness of a dried sample. Therefore, I consider cold stratification as a specialized treatment that places seeds in a particular physiological state, one that retains some of the physiological activity found in fresh samples while also acquiring the dryness and long-term low-temperature storage stability of dried samples. Consequently, when conducting germination experiments, appropriate rehydration treatment is necessary. Based on these considerations, I contend that stratified seeds do not strictly fall into the categories of either fresh or dried samples.
Point 4: Indicate the proper unit for all enzymes
Response 4: We sincerely thanks for your careful reading, we agree with this comment. As suggested by you, we have clarified that the activities of the three enzymes involved in this study were all calculated based on the sample mass, and this has been elaborated in detail in the article. The activity units of all three enzymes remain consistent (U g -1), ensuring the consistency and comparability of the data.
Point 5: There are not so many hormones, but you are writing about hormone and derivatives or precursors - so molecules of hormonal action - it makes the difference.
Response 5: Thank you for your thorough review of the article. As suggested by you, we have changed the “plant hormones” to “metabolites”. You can find the modifications I have made on page 8, paragraph 1, line 353 of the revised manuscript I submitted.
Point 6: Add abbreviation list to all metabolites
Response 6: Thank you for your thorough review of the article. As suggested by you, we have added the Table named “Table 1. All metabolites detected from 15 samples”. It includes the full name, abbreviation and classification of metabolites. You can find the modifications I have made on page 10-11 of the revised manuscript I submitted.
Point 7: suggestion is to avoid we... etc, use rather was hypothesized... etc. Please modify in the rest of the text
Response 7: We sincerely thanks for your careful reading, we agree with this comment. As suggested by you, we have changed the “We hypothesize” to “Hypothesized”. All analogous phrases throughout the entire article have been revised. You can find the modifications I have made on page 11, line 392 and 400; page 16, line 536; page 18, line 574 of the revised manuscript I submitted.
Point 8: Spelling error
Response 8: We sincerely thanks for your careful reading, we agree with this comment. As corrected by you, we have changed the “Analyze” to “Analysis”. You can find the modifications I have made on page 13, line 442 of the revised manuscript I submitted.
Point 9: Uniform the order of red and green drops at fig. A-D
Response 9: Thank you for your thorough review of the article. we have uniformed the drops red (Up) to green (Down). You can find the modifications I have made on page 14, line 481-482 of the revised manuscript I submitted.
Point 10: Do not use this phrase all over the text
Response 10: Thank you for your careful review and suggestions. I have deleted such expressions throughout the text. Specifically, you can find the modifications I have made on page 15, line 498, 505 and 517; page 16, line 526; page 19, line 625 of the revised manuscript I submitted.
Point 11: “Based on these conditions, we speculate that the significant enrichment of the "Circadian rhythm-plant" pathway may represent a mechanism for the seeds to adapt to the stratification environment by regulating their circadian rhythm.” and “We hypothesize that the high enrichment of the EPS pathway may be a physiological response of the seeds to adapt to the stratification environment (low temperature, hypoxia, and low light). On the one hand, the increase in EPS may help enhance the stress resistance of the seeds, protecting them from adverse environmental conditions” please explain what you mean? It is general information.
Response 11: Thank you for your careful review. During the KEGG functional annotation analysis of DEGs between ST1 and ST0 samples, we observed significant enrichment in the "Circadian rhythm plant" and "Exopolysaccharide biosynthesis" pathways. While the enrichment of these metabolic pathways is ubiquitous in plant biology, we have integrated our speculations based on the specific characteristics of our stratified samples, proposing the biological significance behind these enrichment phenomena considering the treatment conditions applied to the samples.
Point 12: The figure legend of Figure 9 is inaccurate.
Response 12: Thank you very much for your careful review of the article. I feel extremely sorry for my previous negligence, and I have made the necessary modifications and improvements. Specifically, I have changed “red and green” to “pink and blue”, you can find the modifications I have made on page 22, line 683 of the revised manuscript I submitted.
